



# Modelling ignition probability for human- and lightning-caused wildfires in Victoria, Australia

Dorph Annalie[1,2], Marshall Erica[1], Parkins Kate A.[1], Penman Trent D.[1]

[1]FLARE Wildfire Research, School of Ecosystem and Forest Sciences, University of Melbourne, Creswick, Victoria, Australia
[2]School of Environmental and Rural Science, University of New England, Armidale, NSW, Australia

*Correspondence to*: Annalie Dorph (annalie.dorph@une.edu.au)

**Abstract.** Wildfires pose a significant risk to people and property which is expected to grow with urban expansion into fire-prone landscapes and climate change causing increases in fire extent, severity and frequency. Identifying spatial patterns associated with wildfire activity is important for assessing the potential impacts of wildfires on human life, property and other

values. Here, we model the probability of fire ignitions in vegetation across Victoria, Australia to determine the key drivers of human- and lightning-caused wildfire ignitions. In particular, we extend previous research to consider the role fuel moisture has in predicting ignition probability while accounting for environmental and local conditions previously identified as important. We used Random Forests to test the effect of variables measuring infrastructure, topography, climate, fuel and soil moisture, fire history, and local weather conditions to investigate what factors drove ignition probability for human- and

lightning-caused ignitions. Human-caused ignitions were predominantly influenced by measures of infrastructure and local weather. Lightning-sourced ignitions were driven by fuel moisture, average annual rainfall and local weather. Both human- and lightning-caused ignitions were influenced by dead fuel moisture with ignitions more likely to occur when dead fuel moisture dropped below 20%. In future, these models of ignition probability may be used to produce spatial likelihood maps which will improve our models of future wildfire risk and enable land managers to better allocate resources to areas of

increased fire risk during the fire season.

## 1 Introduction

Wildfires present a significant risk to both people and property, with this risk increasing as urban areas continue to expand into fire-prone landscapes (Syphard et al., 2013). Wildfire associated risks are likely to increase further with future climate change scenarios predicting increases to fire extent, severity and frequency in fire-prone ecosystems (Bowman et al., 2009; Flannigan

et al., 2009). Wildfires require four key factors to start: sufficient biomass, fuel moisture low enough to allow combustion, weather conditions conducive to fire spread and an ignition source (Archibald et al., 2009; Bradstock, 2010). These factors vary in space and time to influence the risk of a wildfire occurring on any day in a particular location. While weather variables are considered to be determinants of fire size (Bradstock et al., 2014; Penman et al., 2013), the spatial pattern of fire is better



predicted by ignitions and fuels (Parisien et al., 2010; Pausas and Paula, 2012). Understanding spatial patterns in fire activity

is important for assessing the risks and associated impacts of wildfires to human life, property and other values.

Spatial variation in ignition likelihood has been documented in a number of studies, with different patterns observed depending on the ignition source under examination (Bar Massada et al., 2013; Clarke et al., 2019; Liu et al., 2012). Sources of wildfire ignition can either be human-caused or natural. Human-caused ignitions may be either intentional or accidental and are often

related to indicators of human settlement, such as the distance to the nearest road or housing density (Bar Massada et al., 2013; Clarke et al., 2019). Natural ignition sources include lightning strikes, which can account for up to 90% of recorded wildfire ignitions (Clarke et al., 2019; Keeley and Syphard, 2018). The spatial pattern of lightning ignitions differs from human-caused ignitions (Bar Massada et al., 2013) with variables such as local weather or topography driving the probability of lightning ignition (Clarke et al., 2019; Liu et al., 2012). Further, lightning storms may be responsible for multiple ignitions at a time,

potentially resulting in more concurrent fires or larger fires when these ignitions converge (Read et al., 2018). The proximity of human-caused ignitions to population dense areas means early detection of fires is common. However, the potential for lightning-caused ignitions to occur in remote or hard to access locations means detection of these fires can be difficult. Improved models of lightning ignition probability may therefore aid the early detection of and response to wildfires.

Fuels are composed of both live and dead vegetation. Fuel moisture is a critical factor affecting how fire interacts and moves through fuels (Chandler et al., 1983). Fuel moisture in both live and dead fuel contributes to fire ignition, spread and severity in several ecosystems (Chuvieco et al., 2004, 2009; Dennison et al., 2009; Nolan et al., 2016a). For example, fuel moisture content thresholds have been associated with wildfire occurrence in Australian forests and woodlands (Nolan et al., 2016a). Forest fuel moisture content can shift across these thresholds in very short periods of time (e.g. within weeks; Dennison et al.

2009; Nolan et al. 2016a), causing a forest to shift from low to high flammability rapidly. It is therefore important to consider the influence of fuel moisture on fire ignitions at fine temporal scales.

Fuel moisture has previously only been tested in studies of fire ignitions, based on calculations from meteorological data (Dowdy and Mills, 2012; Liu et al., 2012; Miranda et al., 2012; Wotton and Martell, 2005). These methods may lead to

uncertain estimates of fuel moisture in areas with highly heterogeneous topography and vegetation (Nieto et al., 2010). The recent development of fuel moisture estimates at regular temporal intervals from remotely-sensed MODIS data may provide a solution (Nolan et al., 2016a, 2016b). These methods integrate remotely-sensed data with climate modelling inputs to generate layers estimating both live and dead fuel moisture across large spatial and temporal scales. Using these inputs as predictors of fire ignitions may improve our estimates of ignition probability for both human-caused and lightning ignited fires.


Wildfires in south-eastern Australia have resulted in significant loss of human life and property (Blanchi et al., 2010; Filkov et al., 2020; Haynes et al., 2010). A better understanding of both human-caused and lightning-caused ignitions and the



associated risks to human life and property is therefore important for this area. This project aimed to model the probability of fire ignitions across Victoria, Australia. Specifically, we ask are the key drivers of both human-caused and lightning-caused fire ignitions consistent with the global patterns previously reported? We extend previous research of both human and lightning-caused ignitions and ask to what extent does fuel moisture influence ignition probability? And if so, what is the importance of fuel moisture relative to topographic, human and climatic variables?

## 2 Methods

### 2.1 Study Area

Our study area was the state of Victoria in south-eastern Australia. The population in Victoria is ~6.6 million, of which the majority lives in Melbourne ([www.abs.gov.au](www.abs.gov.au), accessed 7th January 2021). Remnant native vegetation in Victoria covers approximately 46% of the state due to previous land clearing for agriculture and human settlements (Fig. 1). There is a climatic gradient across the state with average annual rainfall in the north-west averaging ~300 mm, and in the south-east ranging from 1000 to 1500 mm ([www.bom.gov.au](www.bom.gov.au), accessed 7th January 2021). Average daily maximum temperatures in summer also vary across the state, ranging from 27-30°C in the north-west to 18-24°C in the south-east ([www.bom.gov.au](www.bom.gov.au), accessed 7th January 2021).

### 2.2 Data Compilation

Historical fire ignition data was obtained from the Victorian Country Fire Authority (CFA) and the Department of Environment, Land, Water and Planning (DELWP) for the period between 2000 and 2019 (n = 67,927). These databases have approximately 20 different ignition causes. For this study, ignition causes were reclassified broadly into human-caused (n = 59,146; e.g. from arson or accidental sources) and lightning-ignited fires (n = 8,781) as previous work found consistent patterns in the drivers of the different types of human ignitions in the study area (Clarke et al., 2019).

The analytical pathways for human and lightning ignitions were necessarily different. For the analysis of human-caused ignitions, a set of random points were generated across Victoria from a uniform distribution (n = 75,281). Each random point was assigned a random date and time within the date range of the ignitions data. These random points were used as absence data in the statistical model, providing a random sample of points where fire ignitions did not occur. For the lightning ignitions model, data of all lightning events in Victoria over a certain time period were obtained from the Global Position and Tracking System Pty. Ltd. (GPATS) Australia. These lightning events were each assigned a probability of starting a fire. Therefore, absence data was abundant within the dataset and random points were not required.

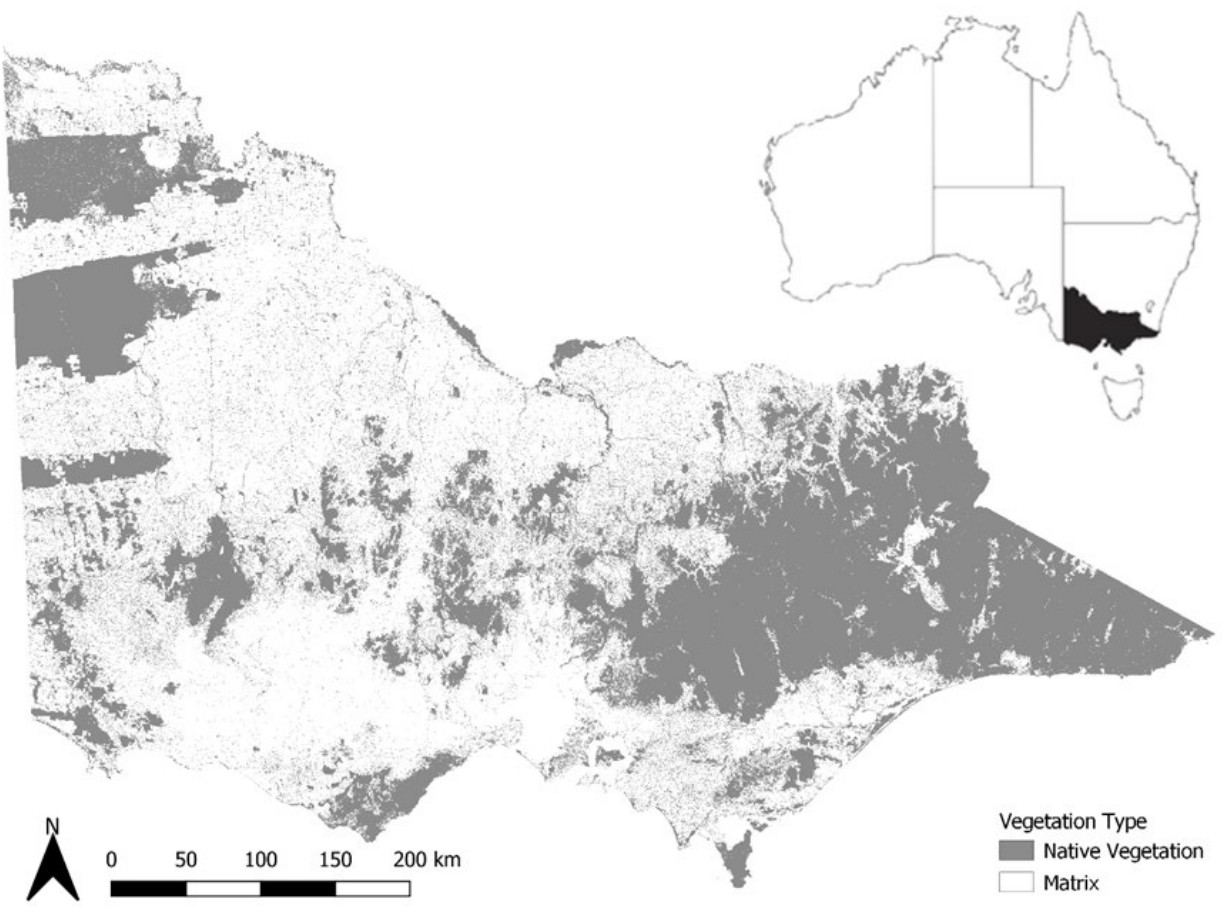

**Figure 1: Study area showing areas of remnant native vegetation and areas of cleared or modified (i.e. matrix) vegetation in Victoria, Australia (source: Native Vegetation Regulation Extent (2017), www.data.vic.gov.au).**


Data from raster layers representing a range of natural and built environments were extracted for each ignition and random point. Details of the different environmental variables used, and descriptions of the layers and their data sources are listed in Table 1. Variables were selected based on those identified as being important from previous studies. For example, human-caused ignitions have a strong relationship with infrastructure variables such as distance to the nearest road and housing density or distance to the nearest settlement (Catry et al., 2009; Clarke et al., 2019; Miranda et al., 2012). Local weather variables, topography and average annual rainfall have also been shown to have an effect on human-caused ignitions (Catry et al., 2009; Clarke et al., 2019; Collins et al., 2015; Liu et al., 2012). Similarly, studies of lightning-caused fires have shown ignitions to be influenced by variables such as: aspect, slope and topographic position (Collins et al., 2015; Miranda et al., 2012); average annual rainfall (Clarke et al., 2019); fuel moisture indices (Dowdy and Mills, 2012; Liu et al., 2012; Miranda et al., 2012;






Wotton and Martell, 2005); local weather (Clarke et al., 2019; Miranda et al., 2012); soil moisture (Liu et al., 2012); and are

predicted be to influenced by changes to fire fuel loads with the time since last fire (Clarke et al., 2019).

**Table 1: Environmental and human-mediated variables used as predictors in model development. Table provides a description of the variable and the source of the data.**

| Variable | Description | Time Range | Source |
|---|---|---|---|
| **Topography** | | | |
| Elevation (m) | Calculated from 30 m Digital Elevation Model (DEM) | N/A | www.data.vic.gov.au |
| Aspect (degrees) | Calculated from 30 m DEM | | |
| Slope (degrees) | Calculated from 30 m DEM | | |
| Topographic Position Index (TPI) | Calculated from 30 m DEM, combining slope position and landform category. Positive TPI values indicate ridges, negative values indicate valleys, and values near zero represent plains and areas of constant slope | | |
| **Fire** | | | |
| Time Since Fire (years) | Derived from Fire History Maps. TSF was set to 100 for ignitions in areas with no mapped fire history. | Annual 2000-2020 | www.data.vic.gov.au |
| **Infrastructure** | | | |
| Housing Density (houses/ km$^2$) | Calculated from vector files of address locations following Clarke et al. (2019) | N/A | www.data.vic.gov.au |
| Distance to the nearest road (km) | Calculated from vector files of roads following Clarke et al. (2019) | | |
| **Climate** | | | |
| Rainfall (mm) | Mean annual rainfall from Worldclim v2.1. | Average from 1970-2000 | www.worldclim.org |
| **Weather** | | | |
| Forest Fire Danger Index (FFDI) | FFDI calculated from gridded hourly temperature, drought factor and humidity data within the VicClim database | Hourly 2000-2017 | Country Fire Authority (CFA) |
| Wind Speed (km/hour) | Wind speed from gridded hourly data within the VicClim database | | |
| **Dryness** | | | |
| Soil moisture (sm_pct) | Extracted from the root zone soil moisture layer provided by the Australian Landscape Water Balance | Annual 2005-2020 | www.bom.gov.au |
| **Fuel Moisture** | | | |
| Live Fuel Moisture Content | Fuel moisture within vegetation. Calculated following Nolan et al. (2016b). (**N.B.** Live Fuel Moisture was limited to areas of native vegetation.) | Annual 2000-2019 | |
| Dead Fuel Moisture Content | | | |


## 2.3 GPATS Lightning Data

Lightning is an electrical discharge generated when positive and negative charges in clouds separate (Latham and Williams, 2001). A flash of lightning from a cloud to the ground can contain a single or several return strokes. Each stroke can be described by its duration, strength (amplitude or current) and polarity (positive or negative). All strokes within a lightning flash





can either follow the same channel to the ground or have several different terminals if the flash branches near to the ground

(Larjavaara et al., 2005).

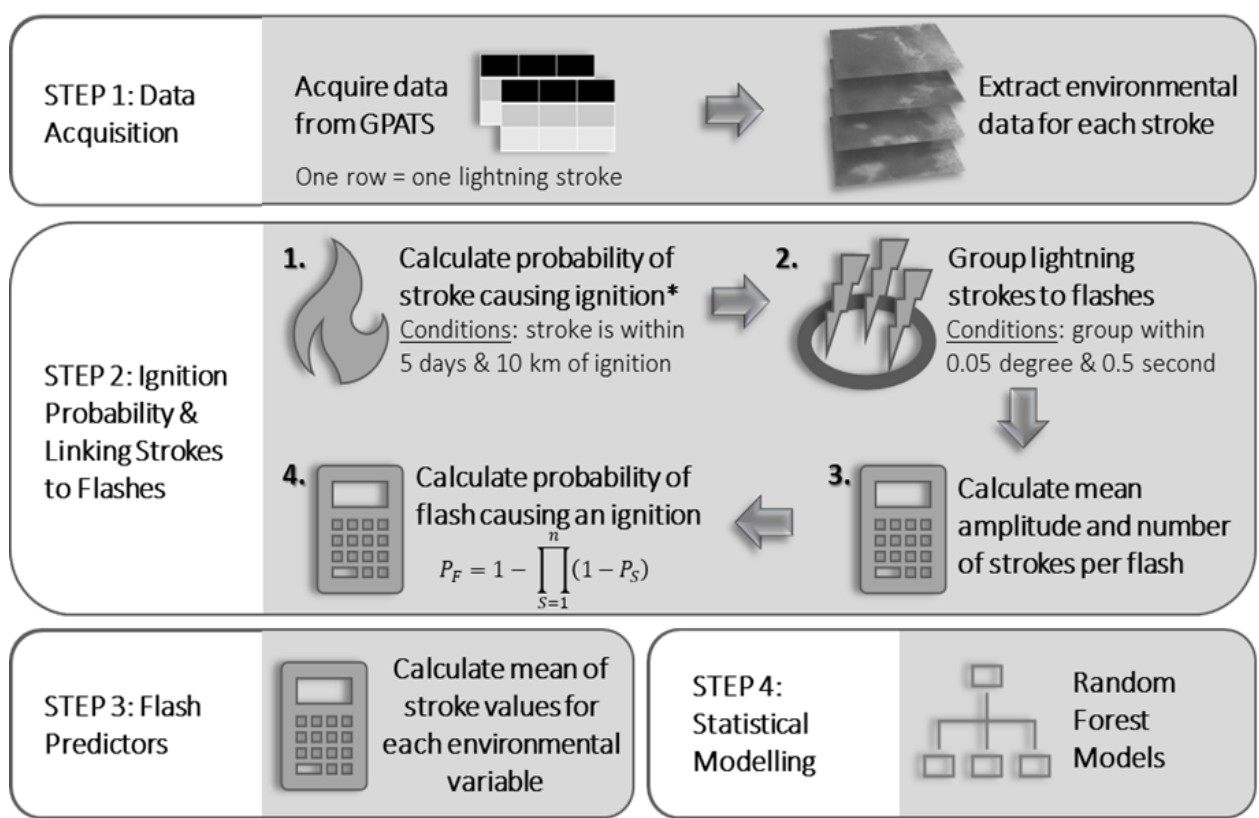

**Figure 2: Lightning data manipulation process. Here a lightning flash is composed of several strokes which can differ in their**
**duration, strength (amplitude), and polarity. Strokes can follow the same channel to the ground or can branch and have several**
**terminals near the ground. *Equations for the calculation of stroke ignition probability are described in text following the method**
**outlines in *(Larjavaara et al., 2005).***

Data of lightning strokes were obtained from Global Position and Tracking System Pty. Ltd. (GPATS) Australia. GPATS uses
triangulation of data from a network of radio receivers to determine the time and location of individual lightning strokes. This

technique distinguishes between cloud-to-cloud strokes and cloud-to-ground strokes and detects the multiple strokes that can

occur within a single lightning flash. The data contains information about the strength of each lightning stroke (amplitude), its

polarity, its time and location. There is some variation within the detection efficiency of the GPATS data due to spatial and

temporal variation in the systems used (Dowdy et al., 2017). The GPATS data obtained covered the state of Victoria from the
period 2004 to 2019 (number of strokes = 3,977,126). The data was simplified for the analysis following the process outlined

in Fig. 2.



The time and location of each stroke was used to extract information from raster layers representing natural and human built variables (Table 1). An ignition probability was calculated for each stroke following the method outlined in Larjavaara et al.

(2005) based on the temporal and spatial proximity of a stroke to an ignition caused by lightning. First the proximity index ($A$) was calculated for each stroke within 10 km of the ignition and within five days preceding the ignition. Approximately 75% of fire ignitions from lightning are detected within three days of the lightning occurring (Wotton and Martell, 2005). We allowed for five days to cover this period and ensure we captured the majority of ignitions. The proximity index was calculated following Eq. (1).

$$A = (1 - T/120))(1 - S/10), \tag{1}$$

where, $T$ is the delay in hours between the time of the stroke to the time of the ignition and S is the spatial distance in kilometres between the stroke and the fire.

The proximity index was then used to calculate ignition probability for each stroke ($P_S$) by dividing the proximity of each

stroke ($A_S$) by the sum of all strokes ($A_i$) within 10 km and five days of the ignition following Eq. (2).

$$P_S = A_S / \sum_{i=1}^{n} A_i, \tag{2}$$

Lightning strokes were grouped into lightning flashes if they occurred within 0.5 seconds and 0.05° in both latitude and longitude. This reduced the size of the dataset (n = 1,994,918) and allowed inclusion of flash multiplicity (a potential indicator

of ignition likelihood; Flannigan and Wotton 1991; Larjavaara et al. 2005) as a predictor in the statistical models. The ignition probability of a flash ($P_F$) was then calculated following the inclusion-exclusion principle described in Eq. (3).

$$P_F = 1 - \prod_{S=1}^{n}(1 - P_S), \tag{3}$$

The inclusion-exclusion principle was used rather than the sum of probabilities as the latter could be greater than one when a lightning stroke was linked to more than one fire. More information on the inclusion-exclusion principle is included in

Appendix 1. Finally, for each flash the environmental data (listed in Table 1) was calculated by averaging the data extracted for each stroke.

## 2.4 Random Forest Modelling

Random Forests were used to determine the probability of an ignition occurring, with separate models built for lighting- and human-caused ignitions. Random Forests are a non-parametric modelling technique with a higher classification accuracy and

reduced risk of overfitting the data compared to other parametric modelling techniques (Breiman, 2001; Cutler et al., 2007). Both classification and regression trees are used in random forests, which are built using a random subset of the data (usually 70%; termed out of bag (OOB) samples). Trees are then ensembled to calculate either the majority vote (classification) or



average value (regression) of predictions in the model (Breiman, 2001). Model accuracy is calculated by comparing the model built on the OOB samples to the data withheld during model development and averaging this across all observations (Cutler

et al., 2007). For classification trees this results in an estimate of classification error, while for regression this gives a measure of variance explained and mean square error.

Variable importance is calculated following two different methods for the two types of Random Forest. For classification trees, variable importance is calculated by summing the decrease in Gini impurity that occurs every time a variable is chosen to split

a node in the classification tree, giving a measure of Mean Gini Index (Cutler et al., 2007). For regression trees, variable importance in calculated by measuring the total decrease in the residual sum of squares that occurs every time a variable is chosen to split a node in a regression tree, giving a measure on Included Node Purity.

Random Forests were used to model lightning- and human-caused ignitions separately. Each ignitions dataset was split again

into those occurring within remnant native vegetation (hereafter "native forest") and those on cleared or modified land (hereafter "matrix") (based on the native vegetation layer from data.vic.gov.au). Models were therefore prepared for four different ignition datasets – human ignitions in native vegetation, human ignitions in matrix, lightning ignitions in native vegetation and lightning ignitions in matrix. Splitting the data into native forest and matrix vegetation was undertaken to allow the inclusion of Live Fuel Moisture as a predictor in models of native forest and also resulted in much reduced computation

times for the models. Live Fuel Moisture has only been modelled for native forest in the south-east of Australia meaning no data was available in matrix areas. As previous work has indicated live fuel moisture thresholds can determine fire activity (e.g. Dennison et al., 2009) we wanted to determine its importance relative to other predictors in this area.

### 2.4.1 Human-caused Ignitions Model Development

A classification Random Forest for human ignitions was built on the ignition data (presence) and a set of random points

(absence).  As the number of points in the presence and absence data were uneven, we used a down sampling method to balance the two classes (Valavi et al., 2021). Down sampling is a method which takes subsamples of the data (with replacement) at each tree so that the classes are equal in sample size. The subsamples are replaced and resampled for every tree that is built. Models were built using all variables listed in Table 1. Soil moisture was removed from the models as it did not improve the overall accuracy of these models and it limited the dataset to ignitions from 2005 onwards. Similarly, of the weather variables,

only FFDI was retained in the model as including the other variables did not improve model accuracy. To ensure optimal model fit, model tuning was conducted to determine the number of trees to grow and the number of variables to sample at each split.

Model validation was done using the OOB error described above and a 10-fold cross-validation procedure. The cross-

validation procedure retained a 10% subset of the data as a test set and built the model on the remaining data. By performing



this cross-validation we were able to retain a complete section of the dataset to validate the model and account for any large variation present in the dataset. The model fit was assessed by comparing the training and testing errors produced by the two model validation procedures. Partial dependence plots (PDPs) were used to examine the relationship of each predictor variable with the probability of an ignition when the other predictors variables are held constant at their average (Friedman, 2001).

### 2.4.2 Lightning-caused Ignitions Model Development

The lightning dataset was highly skewed towards lightning flashes with no probability of ignition (89% of the dataset). Therefore, this analysis was conducted in two stages. The first stage reclassified the ignition probabilities to determine if a flash had no chance of starting a fire (ignition probability = 0) or if the flash had any chance of starting a fire (ignition probability > 0). The reclassified data was zero inflated, so a classification random forest was run using a down sampling method for class imbalanced data (Valavi et al., 2021), as described above. The second stage of the analysis used a regression random forest on all the lightning flashes with any chance of starting a fire to predict what the probability of ignition would be. For consistency, we used the same variables in both stage one and stage two.

Models were built using all the variables listed in Table 1. Testing of variables was undertaken to determine which weather variables were important to include in the model. The best models were produced when both weather variables listed in Table 1 were used. Use of only FFDI resulted in a drop in the accuracy of models. Additionally, the number of strokes within a flash and the average amplitude of these strokes were used as predictor variables. As with the human ignitions model, model tuning was conducted to determine the optimum number of trees and variables to use at each split. Model validation using OOB and cross-validation was conducted for both stages of the lightning model, and partial dependence plots were built to show the predictors effect on the probability of lightning causing an ignition.

All analyses were conducted in R v3.6.3 using the "randomForest" package to build Random Forest models (Liaw and Wiener, 2002). Partial dependence plots were produced using the "pdp" package (Greenwell, 2017) and all graphs were produced in the package "ggplot2" (Wickham, 2009).

### 3. Results

Models of ignition probability from human-caused sources used data from 59,984 ignitions in native forest and 94,034 ignitions in cleared or modified land (i.e. matrix areas). The classification model generated for each of these areas was very accurate with both models predicting between 86.4% and 90.3% of ignitions and non-ignitions (i.e. random points) correctly (Table 2). Models of ignition probability from lightning sources used data from 888,604 flash events in native forest and 986,777 flash events in the matrix. Of these flashes, 11% occurred within 10 km and 5 days of a recorded lightning fire event and so had an ignition probability calculated (average = 3%). The first stage of the lightning modelling process (classification procedure)





recorded all flashes with any probability of ignition as an ignition presence. These models performed well with low error classifying lightning flashes as either likely to start a fire, or with no chance of starting a fire (Table 2). The second stage of the lightning modelling process (regression procedure) used only flashes with a probability of starting a fire. This resulted in

a reduced dataset containing only 77,193 lightning flash events in native forest and 82,762 flashes in matrix. The ability of these regression models to accurately predict the probability that a fire was started from one of the lightning flashes was quite low (~15%; Table 2).

**Table 2: Random Forest results for human ignitions and GPATS lightning data. Model parameters indicates where the random forest was classification or a regression, how many trees were built (*ntree*) and the number of variables tested at each split (*mtry*). Average and standard deviation from the 10-fold cross-validation of classification error for points classed as ignitions compared to points with no ignition are given for classification models. Average and standard deviations from the 10-fold cross-validation are provided for the variance explained ($R^2$) of regression models.**

| Model | Model Parameters | | | Classification Error | | $R^2$ |
|---|---|---|---|---|---|---|
| | *Type* | *ntree* | *mtry* | *Ignition* | *No Ignition* | |
| Human Matrix | Classification | 500 | 2 | $0.100 \pm 0.004$ | $0.101 \pm 0.004$ | |
| Human Native | Classification | 500 | 2 | $0.136 \pm 0.007$ | $0.097 \pm 0.006$ | |
| Lightning Matrix | Classification | 500 | 6 | $0.173 \pm 0.005$ | $0.127 \pm 0.001$ | |
| Lightning Native | Classification | 500 | 6 | $0.226 \pm 0.006$ | $0.131 \pm 0.002$ | |
| Lightning Matrix | Regression | 600 | 4 | | | $15.041 \pm 1.269$ |
| Lightning Native | Regression | 600 | 4 | | | $15.358 \pm 0.929$ |

**3.1 Human-caused Ignitions**

The human ignitions models for both native forest and matrix were predominantly driven by variables measuring human infrastructure: Distance to Road and Housing Density (Fig. 3 a, b). The probability of ignition in both models decreased rapidly as the Distance from Road increased to ~500 m then levelled off (Fig. 4). This effect was not as large within the native forest model as it was in the matrix. Conversely, the probability of ignition increased rapidly in both models as Housing Density

increased to ~100 and remained high above this threshold. FFDI was the next most important variable having a greater influence on the probability of ignition in native forests than in matrix (Fig. 3 a, b). In both models, the probability of an ignition increased rapidly up to an FFDI of ~30 and remained high above this point (Fig. 4). Dead Fuel Moisture and Rainfall had a weaker influence in both models (Fig. 3 a, b). However, both showed thresholds of influence with a lower probability of ignition in parts of the state with low average annual rainfall (<1000 mm) and higher probability of ignition in areas with Dead

Fuel Moisture content below 20 (Fig. 4).



**Figure 3: Variable importance of predictors for human-caused ignitions in (a) matrix and (b) native forest, and lightning ignitions in (c) matrix and (d) native forest. Average and standard deviations of the Mean Decrease in Gini Index for the models produced during the 10-fold cross-validation are shown.**


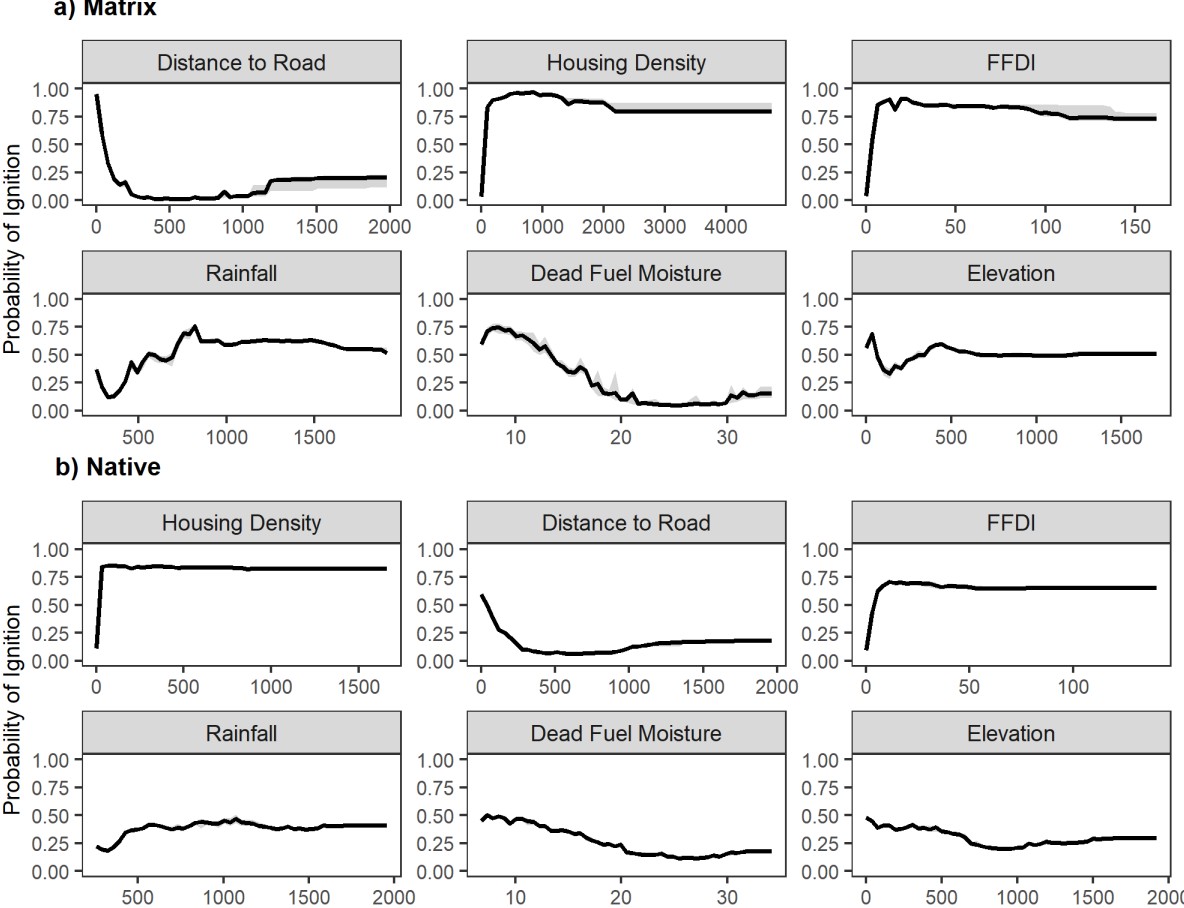

**Figure 4: Partial dependence plot for the six top variables in the random forest model for human-caused fire ignitions in (a) matrix and (b) native forest. Variables are plotted in order of importance. Black lines are the average probability of ignition from the 10 models produced during the 10-fold cross-validation. Grey error bars represent the upper and lower estimates from the 10-fold cross-validation.**

### 3.2 Lightning-caused Ignitions

In classification models predicting whether a lightning flash had any chance of starting a fire, fuel moisture, average annual rainfall, weather and soil moisture were the most important variables (Fig. 3). Dead Fuel Moisture was very important to both models in native forest and in the matrix with a higher probability of ignition when dead fuel moisture was below 20 (Fig. 5). In matrix vegetation, Rainfall was the second most important variable (Fig. 3 c), with the probability of an ignition in parts of the state with average annual rainfall above 1000 mm (Fig. 5 a). Rainfall was less important in native forest (Fig. 3 d), but followed a similar trend with the probability of an ignition increasing as average annual rainfall increased (Fig. 5 b). FFDI was the most important variable in the native forest model (Fig. 3 d), with the probability of an ignition increasing to an FFDI of

~30 and remaining stable at values above this threshold (Fig. 5 b). For FFDI in matrix, the probability of ignitions followed the same trend (Fig. 5 a). Soil moisture was of similar importance to both matrix and native vegetation models (Fig. 3). However, there was no strong trend in either vegetation type, with slightly higher ignition probability in areas with soil moisture below 0.25 (Fig. 5).

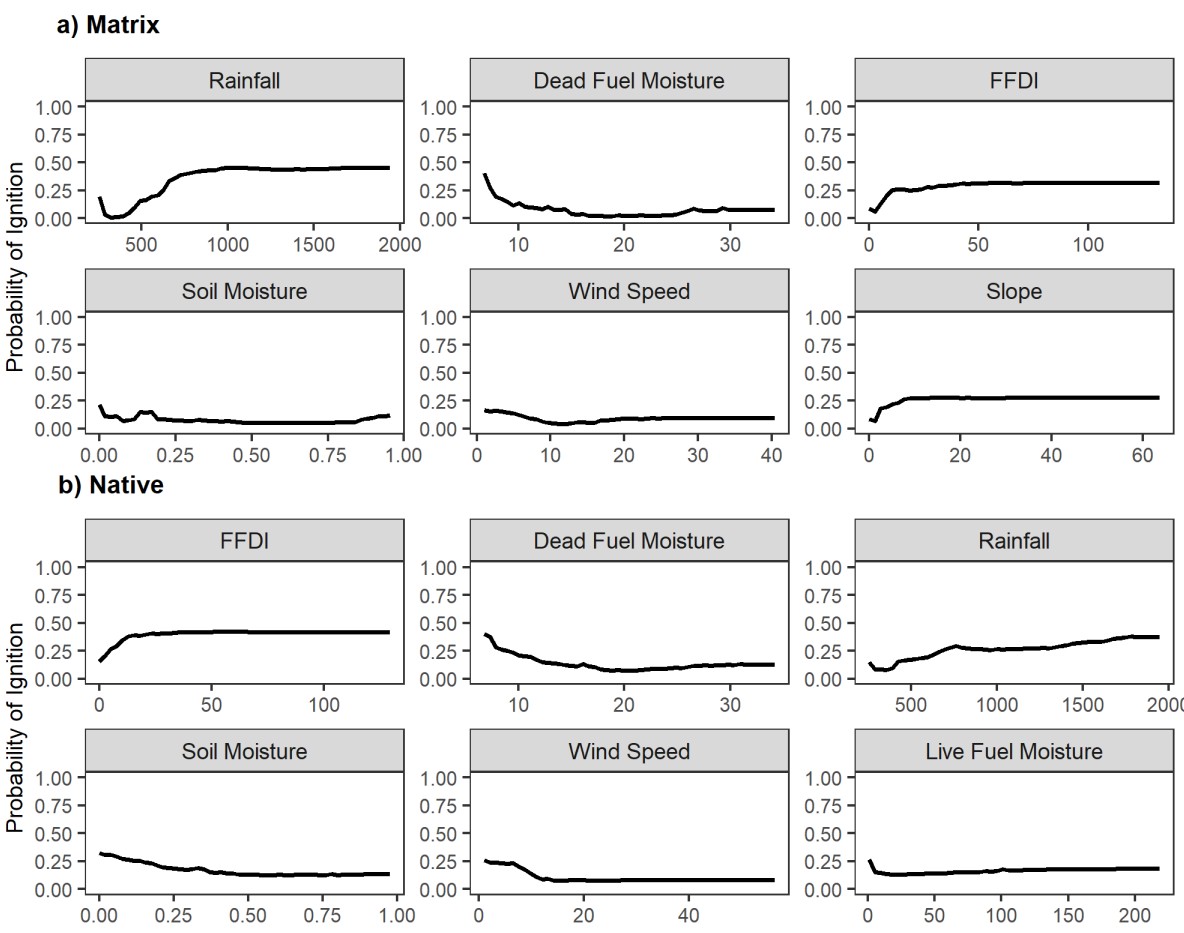

**Figure 5: Partial dependence plot for the six top predictor variables in the random forest models predicting whether a lightning flash has any probability of starting a fire in (a) matrix and (b) native forest. Variables are plotted in order of importance. Black lines represent the average probability of ignition from the 10 models used in the cross-validation. Grey error bars represent the upper and lower estimates from the 10-fold cross-validation.**


    Fuel moisture, lightning stroke amplitude, weather and average annual rainfall were the most important predictors of ignition probability in the regression models for lightning ignition. Dead Fuel Moisture and Average Stroke Amplitude had the strongest effect in both native and matrix vegetation, followed by FFDI and Rainfall (Fig. 6). However, the overall strength of these predictor variables to predict fire ignition probability from a lightning stroke was low (Table 2).


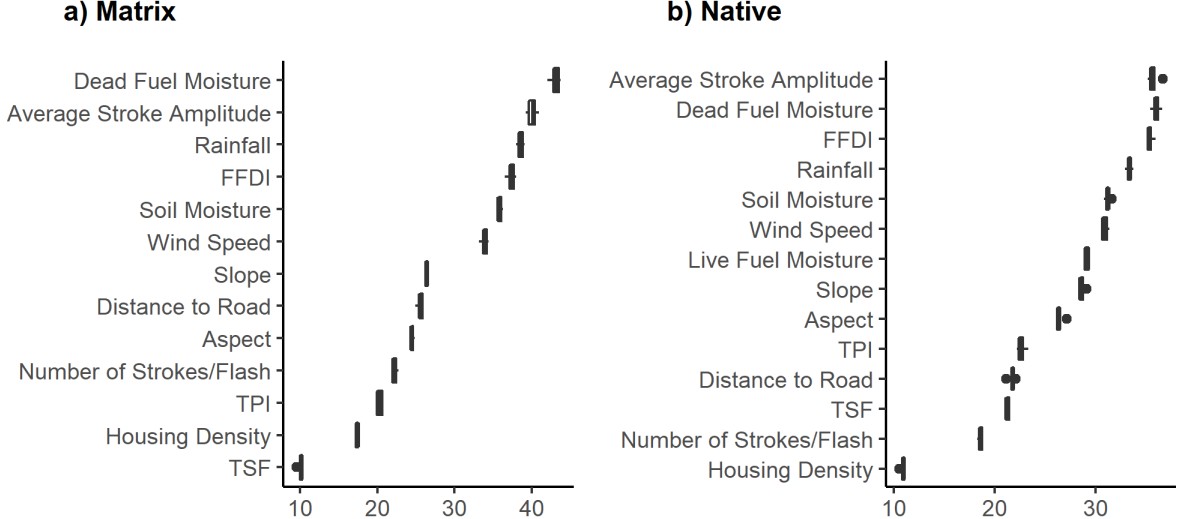

**Figure 6: Variable importance of predictors for the probability of a lightning flash causing an ignition in (a) matrix and (b) native forest. Average and standard deviations of the Included Node Purity calculated by averaging the residual sum of squares over all trees in a random forest model for each model used in the 10-fold cross-validation.**

**4. Discussion**

Understanding spatial patterns of ignition probability is important for the assessment of risks from wildfires to human life, property and environmental values. We evaluated wildfire ignition probability across a broad range of environments in south-eastern Australia to determine key variables driving human-caused and lighting-caused ignitions. We found human-caused ignitions were primarily influenced by human infrastructure and local weather conditions whereas lightning-sourced ignitions

were driven by fuel moisture, average annual rainfall and local weather conditions. In particular, dead fuel moisture influenced ignitions from both sources with ignitions more likely to occur when dead fuel moisture dropped below 20%.

Human-caused ignitions were strongly related to measures of infrastructure and local weather with clear thresholds indicating the influence of these variables. These results are consistent with a number of other studies demonstrating the importance of

infrastructure and weather for predicting spatial patterns in ignitions from human causes (Clarke et al., 2019; Collins et al., 2015; Faivre et al., 2014; Liu et al., 2012; Syphard et al., 2008). These studies also reported similar trends with a higher probability of ignitions closer to roads and housing (Clarke et al., 2019; Faivre et al., 2011; Liu et al., 2012) and a steady increase in ignition probability as FFDI reached 50 (Clarke et al., 2019). The accuracy of the models produced here (86.4% and 90.3%) are also comparable to the accuracy of models in other studies, although differences in modelling approach may

affect interpretation here. For example, Clarke et al. (2019) reported AUCs between 84.6 and 96.7, Collins et al. (2015) reported deviance explained of 83.8% to 88.3% and Liu et al. (2012) reported $R^2$ of 95%. The similarity in predictive accuracies



among these models suggests the inclusion of fuel moisture does not improve overall model performance for human-caused ignitions, despite clear thresholds existing for ignition probability.

Lightning ignition probability was most strongly driven by rainfall, dead fuel moisture and local weather conditions (FFDI). The importance of average annual rainfall and FFDI for lightning ignitions has previously been demonstrated in this region with similar trends in ignitions for these variables reported (Clarke et al., 2019). While studies have previously demonstrated the influence of fuel moisture on lightning ignitions (Dowdy and Mills, 2012; Miranda et al., 2012; Wotton and Martell, 2005), few have done so in conjunction with both local environmental and weather conditions (but see Liu et al., 2012). Incorporation

of variables measuring fuel moisture, environmental factors and local weather may have contributed to a higher prediction accuracy for ignition following lightning flash events using a classification procedure in this study compared to others. Here, model classification accuracy was between 77.4% and 87.3% for models of lightning ignition. Previous studies using different methods and variables to measure classification accuracy have found 67.2% deviance explained (Collins et al., 2015), 73% variability explained (Wotton and Martell, 2005) and an AUC of 79.3 (Clarke et al., 2019). Our estimates may be further

improved by incorporating measures of weather in the days following a lightning event as delays between lightning strike events and fire ignitions are known to occur (Wotton and Martell, 2005).

There is a clear dead fuel moisture threshold effect for both ignition types. Ignition probability was higher in areas where dead fuel moisture was below 20% and dropped almost to zero at fuel moisture ratings above this level. Similar effects of fuel

moisture or surface moisture have been found in previous studies with decreases in ignition probability as moisture content increases for both lightning and human-caused ignitions (Liu et al., 2012; Miranda et al., 2012; Wotton and Martell, 2005). The 20% fuel moisture threshold in dead fuel moisture content mirrors the thresholds found in a study assessing the area burnt by fires in this region (Nolan et al., 2016a), indicating dead fuel moisture levels above 20% significantly reduce both the likelihood of an ignition and the size of a fire. While dead fuel moisture influenced ignition probability in both lightning and

human-caused ignitions, live fuel moisture had only a limited effect in the models and is thus much less likely to determine ignition probability.

Despite the high accuracy achieved in the classification models, lower accuracies were recorded in the regression models of lightning ignition probability. This is likely due partly to discrepancies in the recording locations of both fire ignitions and

GPATS lightning flashes. Within the fire ignitions dataset locations are sometimes recorded at the nearest road or intersection to the fire, rather than the exact latitude and longitude of the ignition. There are also likely to be ignitions missing from the dataset, since many lightning ignitions occurring in remote areas may not be reported or may be reported days after the ignition has occurred. In the GPATs lightning dataset, there is potential recording errors of at least 1 km within key deployment areas (http://www.gpats.com.au/, accessed 25th January 2021) and potentially higher error in remote locations. Additionally, the use

of different recording systems over the years has resulted in spatial and temporal variation in recorded lightning flash locations





(Dowdy et al., 2017). These spatial and temporal discrepancies have likely contributed to the error in the regression lightning model, but may have also influenced classification accuracy in the lightning classification model.

Models of ignition probability were able to be produced with high predictive accuracy despite the spatial error present within
the predictor datasets. These models allow for spatial likelihood maps of ignition probability to be produced at a daily temporal resolution. High accuracy ignition probability models also have application in the estimation of areas where ignition probability may increase under different climate change scenarios. In turn, this would provide better information in models of future fire behaviour and risk in fire-prone landscapes. Further, the ignition probability models developed here use daily weather variables as inputs. This means finer-scale, daily ignition probability maps could be produced on a regular basis. These could inform the
placement of suppression resources for rapid attack when fires occur and also provide better information for the community, such as providing warnings on high fire danger days.

## 6. Conclusion

Globally consistent patterns were found in the drivers of both human and lightning caused ignitions. Human-caused ignitions are predominantly pre-determined by the proximity to human settlement and weather conditions while lightning-caused
ignitions are driven by fuel moisture, average annual rainfall and local weather. Relationships with remotely sensed values of fuel moisture provide a means for better understanding and predicting the likelihood of fires across the landscape on a daily time step. These high accuracy spatial and temporal likelihood maps of ignition probability will improve our models of wildfire risk and enable land managers to better allocate resources during the fire season.



## Appendix 1


Inclusion-exclusion formula for any two events:

$$P(A \cup B) = P(A) + P(B) - P(A \cap B)$$

Where

$$P(A \cap B) = P(A) * P(B)$$

And

$$P(A) = 1 - P(\bar{A}) \text{ and conversely } P(\bar{A}) = 1 - P(A)$$
$$P(B) = 1 - P(\bar{B}) \text{ and conversely } P(\bar{B}) = 1 - P(B)$$

Therefore, the probability of an event not occurring:

$$P(\bar{A}) * P(\bar{B}) = (1 - P(\bar{A})) * (1 - P(\bar{B}))$$

becomes,

$$P(\bar{A}) * P(\bar{B}) = 1 - P(A) - P(B) + P(A \cap B)$$

And conversely, the probability of an event occurring:

$$P(A \cup B) = 1 - (1 - P(A)) * (1 - P(B))$$

Therefore, the inclusion-exclusion formula for the case when there are $n$ number of events becomes:


$$1 - \prod_{i=1}^{n}(1 - P_i)$$

## Data Availability

All data processed could not be made publicly available. For access, the first author can be contacted by email: annalie.dorph@une.edu.au

## Author Contribution

Conceptualization: Annalie Dorph, Trent Penman; Data Curation: Annalie Dorph; Formal Analysis: Annalie Dorph, Erica Marshall; Funding Acquisition: Trent Penman; Investigation: Annalie Dorph, Kate Parkins; Methodology: Annalie Dorph; Project Administration: Annalie Dorph, Trent Penman, Kate Parkins; Supervision: Trent Penman, Kate Parkins; Validation: Annalie Dorph; Visualization: Annalie Dorph; Writing – Original Draft: Annalie Dorph, Kate Parkins, Trent Penman, Erica Marshall; Writing – review & editing: Annalie Dorph, Kate Parkins, Trent Penman, Erica Marshall.



**Competing Interests**

The authors declare that this work was funded by the forest fire risk assessment unit of the Victorian Department of Environment, Land, Water and Planning.

**Acknowledgements**

The fire ignitions data were made available for this study by the Victorian Department of Environment, Land, Water and
Planning. Lightning strike data were made available from the Global Position and Tracking System Pty. Ltd. Australia. The authors would like to thank Stephen Deutsch, Frazer Wilson and Estrella Melero-Blanca for their thoughts and contributions to the development of the analytical approach. We would like to acknowledge the Traditional Owners of the land where this research took place, the Dja Dja Wurrung, Wadawurrung, Wurundjeri, and Widajeri peoples. This research was funded by the Department of Environment, Land, Water and Planning.




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
