# Peer review of "Modelling ignition probability for human- and lightning-caused wildfires in Victoria, Australia"

_Natural Hazards and Earth System Sciences, 2022_

## Author Response (AR1)

Reviewer 1

We thank the reviewer for their comments. We have addressed each of their comments and proposed solutions or clarifications were necessary. Below we outline the changes that have been made in response to each comment (reviewer comments are bold and italicized, our response comments are in plain text).

*I would like to congratulate to the authors for their work.*

*In this work authors train 6 ML algorithm (random forest) using variables of topography, fire, infrastructure, climate, weather, dryness and fuel moisture, to estimate the probability of fire ignition. They create a training dataset for 4 different combinations human/lightning in native/matrix vegetation. The fire ignition data is obtained from Victorian Country Fire Authority, which uses 20 classes for causes of ignition. Then the authors reclassify the ignitions between human or lightning to create the training dataset for the 6 models. In the lightning case, there is the need to create add random fires to create absence data. Then the authors analyse the classification error and the R2. Following, the analyse the importance of the variables for the different models using the decrease of the gini index.*

*Notes:*

*1. It is hard to follow the number of presences and absences for the 4 datasets created for the models. But would good to visualise the presence/absences in Fig 1 for the 4 datasets. Also, the number of presences and absences of each of the 4 datasets.*

*2. Since part the data is not public the previous point 1 would help a bit. Please, specify which part of dataset is not public. Some efforts to show partially that data would help the reader.*

*3. It may be impossible to reproduce the work because the data used is not public.*

*4. L85 "Each random point[...]. These random points" I consider there is in assumption about completeness of CFA data, which it may be fine in this case. But the reader has to accept the assumption without being able to see and check the data.*

Comments 1-4 refer predominantly to the datasets not being publicly available and consequently impacting the reproducibility of the work. The data can be acquired by other researchers; however, it requires gaining permission from the fire agencies who manage the fire ignitions data and purchasing of the lightning data from GPATS.

As suggested by the reviewer we have changed Figure 1 to a four-panel figure plotting: (a) human ignitions (human presence data); (b) randomly generated points (human absence data); (c) recorded lightning ignitions; and (d) all recorded lightning strikes. All graphs used point locations for the data except, lightning strike locations where a density map was required as there were too many occurrences to represent using point data.

*5. When merging all these datasets would be good to have quick look to the temporal and spatial resolution (in table 1?).*

We have added rows to the end of Table 1 with the response variables, a description of the data and its spatial resolution and the temporal scale covered as suggested.

***6. It is not so trivial for the reader to do a diagram of the use of the data flow for the lightning models. This point is more a doubt than nothing else. So, data from GPATS is used to create a training dataset computing the probability of ignition, and then, applying a two stage approach (classification and after regression) to avoid 0 probability of ignition. But, when the fire ignitions (caused by lightning) from CFA are used? GPATS give you lightning info occurrences, you use that to model the probability of fire ignition (Larjavaara et al 2005) as a "ground truth", and after evaluate that with the two stage approach with the variables from table 1. What about the presences of ignition by lightning from CFA? How the model performs with these cases? So, here you are comparing two models (Larjavaara vs two stage random forest) using considering GPATS for Larjavaara and the data from table 1 for the two stage random forest. However, the 8781 cases from CFA seems not to be used at all. I assume the contribution here is the comparison of two completely different models that use different input data. Would be possible to see just the performance for that 8781 cases for matrix and native.***

There a couple of points here where we think the reviewer has misunderstood the process illustrated in the flow chart. The diagram attempts to highlight that once GPATS and CFA data was acquired, the equations from Larjavaara were used to link the two datasets by:

- calculating the probability of lightning strokes (GPATS) being linked in space and time to the lightning strike ignitions (CFA & DELWP) and;
- group the lightning strokes (i.e. single cloud to ground transition of electricity) into lightning flash events (i.e. what is seen by the eye when lightning strikes)

When GPATS lightning strikes were within 5 days and 10 km of a CFA ignition then they were counted as ignition presences in the final dataset. Any lightning strokes which were not linked to a CFA or DELWP ignition were counted as absences. Thus, it is not a method of ground-truthing as the reviewer has suggested, but a method to integrate the two datasets to give presence-absence data. Consequently, the CFA and DELWP data is not compared separately, because it is integrated into the presence/ absence calculations for the lightning strike data. We considered this approach to provide more accurate information than following the same approach used in the human ignitions model in which the 8781 recorded CFA and DELWP ignitions would be recorded as presences, and absences would be generated using random points. The process of linking lightning strokes to fires and grouping them together is explained in text. We have added further clarification in text to make the outcomes of this process clearer by adding to the end of the section regarding GPATS lightning data after the lines 163 – 166 which reads:

 "The inclusion-exclusion principle was used rather than the sum of probabilities as the latter could be greater than one when a lightning stroke was linked to more than one fire. More information on the inclusion-exclusion principle is included in Appendix 1."

We have added the following:

"By completing these calculations, all lightning strokes that were linked to a CFA or DELWP fire ignition using the proximity index were assigned an ignition probability greater than zero. Any lightning strokes that were not linked to a CFA or DELWP ignition by the proximity index were assigned an ignition probability of zero and were treated as absence data in the analysis."

***7. I would specify the criteria for the reclassified ignition causes from the CFA classes.***

The criteria were specified in the text (see Lines 80 - 83 which read): "For this study, ignition causes were reclassified broadly into human-caused (n = 59,146; e.g. from arson or accidental sources) and

lightning-ignited fires (n = 8,781) as previous work found consistent patterns in the drivers of the different types of human ignitions in the study area (Clarke et al., 2019).". However, to improve clarity further we have added all of the ignition classes to each of the classifications rather than providing an example. The sentence at lines 85-86 now reads:

"For this study, ignition causes were reclassified broadly into human-caused (n = 59,146; from arson or accidental sources including burn offs, campfires, electrical, fireworks, heat or cutting equipment, power transmission lines, re-light, vehicles, and waste disposal) and lightning-ignited fires (n = 8,781) as previous work found consistent patterns in the drivers of the different types of human ignitions in the study area (Clarke et al., 2019)."

***8. One doubt in the partial dependence plot. When the rainfall increases, also increases the probability of ignition. The rainfall of these plots is an annual mean as mentions in table 1. So, each pixel has the average from 1970 to 2020 of the mean annual rainfall? So, each pixel has a single value with the average of the mean of the annual rainfall. It would mostly provide spatial information and is not related at all with seasonality moisture. It may be related with the fuel production. I suggest try to define the parameters of rainfall (time) in the figures...***

In this paper, average annual rainfall was not being used as a measure of seasonal moisture variation. Instead, we represent moisture variability using dead and live fuel moisture estimates. Here, we use average annual rainfall to represent the number of storms different parts of the landscape are subject to. Further, average annual rainfall is linked to fuel load and accumulation (Thomas et al., 2014) and this is likely to be the cause of increased fire ignitions. We have added text to the discussion explaining this (lines 322-323):

"Average annual rainfall is known to influence fuel load and accumulation (Thomas et al., 2014), thus areas with higher average annual rainfall have greater fuel availability and consequently are likely to have an increased probability of ignition."

***9. L265 "In matrix vegetation, Rainfall was the second most important variable (Fig. 3c). In the manuscript I have rainfall in on top for Fig. 3c., second is dead fuel moisture.***

The reviewer is correct, this is mistake in the writing of the manuscript. The sentence should read "In matrix vegetation, Rainfall was the most important variable (Fig. 3c)." This has been amended in text.

***Please cite the datasets used wherever is possible.***

All predictor variable datasets are listed in Table 1. As mentioned above, we have added the response variables to the tables to re-iterate the spatial and temporal information as well as the datasets source.

***The results and conclusions of the work rely on the data. Please describe more the dataset used, versions, resolutions and for the ones that are not public may be analysis and map. See for instance Clark et al. 2019, similar concept, but different methods and inputs. Despite this, the description on the data used is more clear and easy to follow.***

Comments are provided on this above.

***Again, congratulations for the work. It is really interesting and potentially useful****.*

Reviewer 2

We thank the reviewer for their comments. We have addressed each of their comments and proposed solutions or clarifications were necessary. Below we outline the changes that have been made in response to each comment (reviewer comments are bold and italicized, our response comments are in plain text).

***This study models ignition probability in the state of Victoria, Australia, distinguishing between human and natural fires. It is in generally well written, clear, and I see no issues in the methodology. However, some modelling choices in terms of the independent variables chosen are debatable (see specific comments below). The authors select a number of putative independent variables that are expected to influence such likelihoods, but in practical terms the effects of some are unconvincing, as judged from the partial dependence plots. In order words, the models could be more parsimonious. Overall, I felt the Introduction and Discussion sections could be made stronger than currently they are; just a suggestion that if implemented would increase the relevance of the work.***

***Specific comments***

***L28. Not sure about this distinction. Fire size is a feature of spatial pattern. So, what's spatial pattern in the context of the sentence?***

Yes, fire size in a component which can define elements of spatial pattern. However, spatial pattern is more generally considered to be the arrangement of different patches in space and the relationships among them geographically (often considered as spatial composition and configuration). In this context, ignitions and fuels can determine where fires start and spread, affecting spatial patterns fires create over time. To address the reviewers comment and to increase clarity around the wording here we have changed "fire size" to "fire annual extent" (line 28).

***L49. I don't think weeks qualify as very short period of time … The shift across dead fuel moisture thresholds is very often on a daily scale as fires cease to spread at night and resume the next morning.***

This is an error in phrasing on our part. The sentence being referred to in text currently reads: "Forest fuel moisture content can shift across these thresholds in very short periods of time (e.g. within weeks, …)". Instead based on the work it is referring to, the sentence should say "from hours to a week". This has been amended in text (lines 49-50).

***L64-66. Check for better phrasing.***

The reviewer is referring to the sentence: "This project aimed to model the probability of fire ignitions across Victoria, Australia. Specifically, we ask are the key drivers of both human-caused and lightning-caused fire ignitions consistent with the global patterns previously reported? We extend previous research of both human and lightning-caused ignitions and ask to what extent does fuel moisture influence ignition probability"

We have re-phrased the sentence as follows:

"This research aimed to model the probability of fire ignitions across Victoria, Australia. Specifically, we ask whether key drivers of human- and lightning-caused fire ignitions are consistent with previously reported global patterns. In addition, we extend this previous research to determine to what extent fuel moisture influences ignition probability."

*Table 1: add "relative" before "humidity" in the FFDI row.*

Amended as suggested.

*L79. Why was live fuel moisture included as a predictor? All fires start on the dead component of the fuel complex and lightning-caused fires in particularly are highly dependent on the forest floor moisture content. While live m.c. definitely should play a role in fire spread such role has never been quantified or even demonstrated outside the lab. Also, having a model with live m.c. as a predictor can do more harm than good, given the spatial mismatch (in terms of scale) between actual live m.c. and estimated live m.c. and how uncertain remotely-sensed estimates of live m.c. are.*

Live Fuel Moisture was included as it has been linked to fire behaviour and extent (Nolan et al., 2016, 2018), therefore we might expect ignitions to be influenced by live fuel moisture. The spatial mismatch is an issue for live or dead fuel moisture when using the remotely sensed data. Of course, there are fine scale variations in each of these, however when predicting landscape fire risk at a daily scale these values appear to be useful. We therefore wish to retain the analysis as it is.

*I also see that FFDI is not influent beyond a very low value, which is probably an outcome of having both FFDI and dead m.c. in the model – correlation between the two is expected to be strong. Or maybe not, because Nolan's m.c. model (if I recall well) is for 10-hr fuels, not really the fine fuels that drive fire ignition and spread. In this respect, and also having in mind management applications, why did the authors employ Nolan's model in lieu of the m.c. models used by fire management agencies? Both depend of the same variables, i.e. RH and temp.*

All variables which were included in the models were tested for correlations and any combination of variables with a Pearson correlation coefficient above 0.7 was excluded from the model. Further, the threshold effect of FFDI above 20 is a result supported by other published research (Clarke et al., 2019).

The question of which model to use is potentially irrelevant for this research and we argue we should not focus on existing models used by management agencies. Instead, we should be using the most up to date representations of values to ensure we are producing the most accurate models of processes.

*L271. Like in the case of FFDI/dead m.c. I see potential confusion/redundancy with the simultaneous use of soil moisture and the FFDI, as the FFDI includes a drought component that should be correlated with the upper soil moisture.*

It is possible the reviewers' previous understanding of the relationship between soil moisture and drought factor has influenced their interpretation of how this relationship is represented in our data. As mentioned above, all predictor variables were tested for correlations. When Pearson correlation coefficient was > 0.7 for pairs of variables, one of them was removed. For lightning ignitions, FFDI and soil moisture had a -0.4 correlation, well below the threshold used to eliminate variables. The root zone soil moisture variable that was used measures the percentage of available water content in the top 1 of the soil profile. While FFDI does include a component of soil moisture (drought factor), the inclusion of relative humidity, air temperature and wind velocity alters this measure enough that the variables are not correlated.

*L310. What is the plausible explanation for the effect of increased lightning ignition likelihood with higher annual rainfall? Higher NPP and thus higher fuel accumulation?*

Average annual rainfall is known to influence fuel load and accumulation (Thomas et al., 2014) and this is likely to be the cause of increased fire ignitions. We have added text to the discussion explaining this:

"Average annual rainfall is known to influence fuel load and accumulation (Thomas et al., 2014), thus areas with higher average annual rainfall have greater fuel availability and consequently are likely to have an increased probability of ignition."

**References**

Clarke, H., Gibson, R., Cirulis, B., Bradstock, R. A. and Penman, T. D.: Developing and testing models of the drivers of anthropogenic and lightning-caused wildfire ignitions in south-eastern Australia, J. Environ. Manage., 235, 34–41, doi:10.1016/j.jenvman.2019.01.055, 2019.

Nolan, R. H., Boer, M. M., Resco de Dios, V., Caccamo, G. and Bradstock, R. A.: Large-scale, dynamic transformations in fuel moisture drive wildfire activity across southeastern Australia, Geophys. Res. Lett., 43(9), 4229–4238, doi:10.1002/2016GL068614, 2016.

Nolan, R. H., Hedo, J., Arteaga, C., Sugai, T. and Resco de Dios, V.: Physiological drought responses improve predictions of live fuel moisture dynamics in a Mediterranean forest, Agric. For. Meteorol., 263, 417–427, doi:10.1016/J.AGRFORMET.2018.09.011, 2018.

Thomas, P. B., Watson, P. J., Bradstock, R. A., Penman, T. D. and Price, O. F.: Modelling surface fine fuel dynamics across climate gradients in eucalypt forests of south-eastern Australia, Ecography (Cop.)., 37(9), 827–837, doi:10.1111/ECOG.00445, 2014.